# Functional Roles of CD26/DPP4 in Bleomycin-Induced Pulmonary Hypertension Associated with Interstitial Lung Disease

**DOI:** 10.3390/ijms25020748

**Published:** 2024-01-06

**Authors:** Tadasu Okaya, Takeshi Kawasaki, Shun Sato, Yu Koyanagi, Koichiro Tatsumi, Ryo Hatano, Kei Ohnuma, Chikao Morimoto, Yoshitoshi Kasuya, Yoshinori Hasegawa, Osamu Ohara, Takuji Suzuki

**Affiliations:** 1Department of Respirology, Graduate School of Medicine, Chiba University, Chiba 260-8670, Japan; 2Synergy Institute for Futuristic Mucosal Vaccine Research and Development, Chiba University, Chiba 260-8670, Japan; 3Department of Therapy Development and Innovation for Immune Disorders and Cancers, Graduate School of Medicine, Juntendo University, Tokyo 113-8421, Japan; 4Department of Biomedical Science, Graduate School of Medicine, Chiba University, Chiba 260-8670, Japan; 5Department of Applied Genomics, Kazusa DNA Research Institute, Chiba 292-0818, Japan

**Keywords:** CD26, dipeptidyl peptidase-4, pulmonary hypertension, interstitial lung disease

## Abstract

Pulmonary hypertension (PH) with interstitial lung diseases (ILDs) often causes intractable conditions. CD26/Dipeptidyl peptidase-4 (DPP4) is expressed in lung constituent cells and may be related to the pathogenesis of various respiratory diseases. We aimed to clarify the functional roles of CD26/DPP4 in PH-ILD, paying particular attention to vascular smooth muscle cells (SMCs). *Dpp4* knockout (*Dpp4*KO) and wild type (WT) mice were administered bleomycin (BLM) intraperitoneally to establish a PH-ILD model. The BLM-induced increase in the right ventricular systolic pressure and the right ventricular hypertrophy observed in WT mice were attenuated in *Dpp4*KO mice. The BLM-induced vascular muscularization in small pulmonary vessels in *Dpp4*KO mice was milder than that in WT mice. The viability of TGFβ-stimulated human pulmonary artery SMCs (hPASMCs) was lowered due to the *DPP4* knockdown with small interfering RNA. According to the results of the transcriptome analysis, upregulated genes in hPASMCs with TGFβ treatment were related to pulmonary vascular SMC proliferation via the Notch, PI3K-Akt, and NFκB signaling pathways. Additionally, *DPP4* knockdown in hPASMCs inhibited the pathways upregulated by TGFβ treatment. These results suggest that genetic deficiency of *Dpp4* protects against BLM-induced PH-ILD by alleviating vascular remodeling, potentially through the exertion of an antiproliferative effect via inhibition of the TGFβ-related pathways in PASMCs.

## 1. Introduction

Pulmonary hypertension (PH) is defined by elevated mean pulmonary arterial pressure (>20 mmHg [1]). Patients with PH often develop right heart failure if no appropriate treatment is administered. PH is classified into five clinical phenotypes according to its pathophysiology. Each phenotypic group requires a different treatment strategy and shows a different prognosis [2]. Among the five phenotypes, the prognosis of patients with pulmonary arterial hypertension (PAH) has remarkably improved in recent years with continuous intravenous prostaglandin I_2_ therapy [3] and upfront combination therapy with pulmonary vasodilators [4].

Group III PH develops secondary to chronic lung diseases such as interstitial lung diseases (ILDs) and chronic obstructive pulmonary disease. A proportion of patients with ILDs comprises those with chronic progressive lung diseases characterized by varying degrees of inflammation and fibrosis in the lung interstitium. Idiopathic pulmonary fibrosis (IPF) is a clinical phenotype of ILDs that often develops into an intractable condition, and approximately 8–15% of IPF cases are associated with PH [5]. In general, the effects of pulmonary vasodilators are limited in PH with ILD (PH-ILD), and patients with PH-ILD often exhibit clinical worsening with pulmonary vasodilators [6,7,8], although inhaled prostaglandin I_2_ therapy does improve exercise tolerance [9]. Therefore, the prognosis of patients with PH-ILDs has remained worse than that of patients with PAH [10], and a better understanding of PH-ILD pathogenesis is necessary to develop new treatment strategies other than using pulmonary vasodilators.

In patients with PH, the main pathological findings include vascular remodeling of the small pulmonary arteries [11], partly due to the persistent pressure overload of the pulmonary circulatory system. They are characterized by thickening of the tunica media accompanied by an increase in pulmonary artery smooth muscle cells (PASMCs) and luminal narrowing with cell proliferation of the intima and fibrosis. Plexiform lesions are sometimes observed as the disease progresses in the final stage of pulmonary circulatory failure. In Group III PH, which encompasses both pulmonary vascular and parenchymal lung lesions, the pathology of pulmonary vascular remodeling differs from that of PAH, and thickening of the tunica media is the primary pathological feature [12].

Bleomycin (BLM) is an anticancer drug used against malignant lymphoma and testicular cancer, which inhibits DNA synthesis and introduces single or double-strand scissions in DNA. However, BLM is known to cause lung fibrosis as a severe adverse effect [13]. Therefore, it is used to establish not only ILD models in mice and rats [14] but also PH-ILD models in mice [15]. Intratracheal administration is common for BLM exposure; however, intravenous, intraperitoneal, and intranasal administration have also been used [14]. Intraperitoneal BLM administration is especially used in establishing a PH-ILD model, assuming uniform vascular remodeling in the lungs of mice.

CD26/Dipeptidyl peptidase-4 (DPP4) is a protein with a molecular weight of 110 kDa. It is expressed as a type II membrane-bound protein on the surface of multiple types of human and rodent cells [16]. The soluble form of DPP4 enters blood circulation after shedding from the cell surface. The enzymatic activity of CD26/DPP4, such as degradation and inactivation of incretins, has been the focus of many studies, and DPP4 inhibitors have been used clinically for the treatment of diabetes mellitus [17]. CD26/DPP4 was originally established as a T-cell activation antigen that participates in immune stimulation [18]. It plays multiple roles, including those in the progression of inflammation and fibrosis in various diseases [19]. DPP4 inhibitors can not only improve respiratory diseases [20], such as acute respiratory distress syndrome [21] and ILDs [22], but also cardiovascular diseases [23], such as left heart dysfunction [24] and atherosclerosis [25].

DPP4 inhibitors could be a new treatment option for patients with PH [26]. These inhibitors may prevent the pathological progression of PH by influencing pulmonary vascular cells and lung fibroblasts. In vivo, sitagliptin, a DPP4 inhibitor, mitigated the elevation of right ventricular systolic pressure (RVSP) in a hypoxia-induced PH rat model and attenuated pulmonary artery remodeling by decreasing the number of PASMCs in the media [27]. Moreover, sitagliptin treatment suppressed PDGF-BB-induced migration of cultured human PASMCs in vitro [28]. Based on the above-mentioned observations, we speculated that CD26/DPP4 plays mechanistic roles in the pathogenesis of PH-ILD. Through this study, we aimed to clarify the functional roles of CD26/DPP4 in PH-ILD using *Dpp4*-deficient mice in a BLM administration model and using DPP4-siRNA in cultured human PASMCs.

## 2. Results

### 2.1. BLM-Induced Pulmonary Hypertension Was Attenuated in Dpp4KO Mice

First, we evaluated the expression levels of CD26/DPP4 in the cellular components of the lungs of wild-type (WT) and *Dpp4* knockout (*Dpp4*KO) mice. Real-time quantitative PCR analysis showed that *Dpp4* expression was significantly lower in *Dpp4*KO mice than that in WT mice (Figure 1a). Flow cytometric analysis also showed that CD26/DPP4 expression was substantially low or nearly absent in *Dpp4*KO mice (Figure 1b,c).

In BLM-administered WT mice, RVSP, cardiac output (CO), and the maximum rate of pressure rise (Max dP/dt) were higher compared with those in PBS-administered WT mice (Figure 1d,f,g), although the difference in heart rate (HR) (Figure 1e) was minor. These hemodynamic changes after the BLM challenge were attenuated in *Dpp4*KO mice (WT/BLM versus *Dpp4*KO/BLM, RVSP: 33.0 versus 26.6 mmHg [*p* < 0.01], CO: 0.98 versus 0.79 [*p* = 0.068], and Max dP/dt: 2.55 versus 1.49 [*p* < 0.05]) (Figure 1d,f,g). Regarding right ventricular hypertrophy, the weight ratio of the right ventricle to the left ventricle plus the ventricular septum (RV/LV + S) and RV/body were greater in WT mice after the BLM challenge. The increase in RV/LV + S was significantly alleviated in *Dpp4*KO mice (WT/BLM versus *Dpp4*KO, 0.39 versus 0.30 [*p* < 0.05]) (Figure 1h,i). No apparent difference in mortality was observed either for WT/PBS and *DPP4*KO/PBS mice or for WT/BLM and *DPP4*KO/BLM mice (Appendix A).

### 2.2. Media Thickening in Small Pulmonary Vessels Was Attenuated in Dpp4KO Mice

The BLM challenge thickened the media of the small pulmonary vessels in WT mice (WT/BLM) compared with that in PBS-treated mice (WT/PBS); however, the media thickness was attenuated in BLM-treated *Dpp4*KO mice (*Dpp4*KO/BLM) (Figure 2a). As evident from the quantitative evaluation of vascular muscularization in small pulmonary arteries, the number of partially or fully muscularized vessels was significantly greater in WT/BLM mice than that in WT/PBS mice (partially muscularized: *p* < 0.001 and fully muscularized: *p* < 0.05). The total number of muscularized vessels (partially and fully) in *Dpp4*KO/BLM mice was significantly lower than that in WT/BLM mice (*p* < 0.05) (Figure 2b), whereas no individual differences were observed between WT/BLM and *Dpp4*KO/BLM (partially muscularized: *p* = 0.29 and fully muscularized: *p* = 0.32) (Figure 2c).

The expression levels of α-SMA in CD31^+^CD45^−^ pulmonary cells (partial endothelial-to-mesenchymal transition cells), evaluated as mean fluorescence intensity (MFI), were higher after the BLM challenge (WT/PBS versus WT/BLM: *p* < 0.05). Notably, the expression levels were significantly lower in *Dpp4*KO/BLM than those in WT/BLM (*p* < 0.05). The expression levels of α-SMA in whole lung cells of WT/BLM were not significantly different from those in *Dpp4*KO/BLM (*p* = 0.36) (Figure 2d,e,f).

The BLM challenge caused fibrosis in the lungs and the right ventricle in both WT and *Dpp4*KO mice (Figure 2g,h). The Ashcroft scale, a quantitative lung fibrosis histological evaluation, in WT/BLM was more severe than that in WT/PBS (*p* < 0.001); however, it was not significantly different between WT/BLM and *Dpp4*KO/BLM (*p* = 0.95) (Figure 2i).

### 2.3. Cell Proliferation and Cytotoxicity Were Reduced by DPP4-siRNA Treatment in Cultured hPASMCs

Expression levels of CD26/DPP4 in cultured human PASMCs (hPASMCs) treated with either control-siRNA (Control) or *DPP4*-siRNA (*DPP4* knockdown, *DPP4*KD) were examined in vitro. Real-time quantitative PCR revealed that the mRNA expression level of *DPP4* was reduced by *DPP4*-siRNA (Figure 3a). Flow cytometric analysis also revealed that the expression levels of CD26/DPP4 were reduced by *DPP4*-siRNA treatment (Figure 3b,c).

Proliferation and cytotoxicity assays were performed to explore the potential role of CD26/DPP4 in the proliferation and cytotoxicity of PASMCs. A proliferation assay of cultured hPASMCs revealed that the enhanced cell viability induced by TGFβ treatment was suppressed by *DPP4*-siRNA treatment (Figure 3d). A cytotoxicity assay revealed that cell damage was unchanged after TGFβ treatment, whereas it was significantly lower in *DPP4*-siRNA-treated cells than that in the control cells (Figure 3e).

### 2.4. Transcriptome Analysis of Cultured hPASMCs after Treatment with TGFβ and DPP4-siRNA

To explore the potential effects of TGFβ treatment on the transcriptome signature of cultured hPASMCs, a comparison between Control/PBS and Control/TGFβ groups was performed. Principal component analysis and heat maps with hierarchical clustering revealed that gene expression patterns differed between the groups (Figure 4a,b). Similarly, to reveal the effects of *DPP4*-siRNA, a comparison between the Control/PBS and *DPP4*KD/PBS groups was carried out (Figure 4c,d). Furthermore, to reveal the effects of *DPP4*-siRNA under TGFβ treatment, a comparison between Control/TGFβ and *DPP4*KD/TGFβ groups was conducted (Figure 4e,f).

Enrichment analysis (gene ontology [GO] and Kyoto Encyclopedia of Genes and Genomes [KEGG] pathways) comparing the Control/PBS and Control/TGFβ groups suggested that TGFβ treatment upregulated genes related to smooth muscle cell (SMC) proliferation and differentiation, growth factor stimulation, further TGFβ production, augmentation of TGFβ and TGFβ receptor responses, and pulmonary vascular SMC proliferation in hPASMCs (Table 1A). As an intervening pathway of TGFβ treatment, the genes related to pathways such as Notch, PI3K-Akt, and NFκB signaling pathways were upregulated according to the KEGG pathway analysis (Table 1B).

DPP4-siRNA treatment seemed to downregulate the genes and pathways that were upregulated by TGFβ treatment (Table 2). Gene expression levels associated with the *DPP4*, the *TGF* family, and the Notch and NFκB pathways were measured; the results are summarized in Table 3 and Figure 4g–r. The expression levels of *TGFBR1* after TGFβ treatment were downregulated by *DPP4*-siRNA (Table 3A and Figure 4i). Additionally, the expression levels of genes related to the Notch3 and NFκB signaling pathways were downregulated by *DPP4*-siRNA (Table 3B,C and Figure 4k–r).

## 3. Discussion

In the present study, we demonstrated that BLM-induced pulmonary vascular remodeling associated with media thickening was attenuated in *Dpp4*KO mice. In vitro experiments showed that the TGFβ-enhanced proliferative capacity of cultured hPASMCs was suppressed by *DPP4*-siRNA treatment. Transcriptome analysis revealed that TGFβ treatment of cultured hPASMCs upregulated genes related to pulmonary vascular SMC proliferation, involving the Notch, PI3K-Akt, and NFκB signaling pathways in cultured hPASMCs. Conversely, application of *DPP4*-siRNA to cultured hPASMCs canceled these TGFβ-induced events. Specifically, *TGFBR1* and genes associated with the Notch3 and NFκB signaling pathways were downregulated by *DPP4*-siRNA. These results suggest that genetic deficiency of *Dpp4* provides protection against BLM-induced PH-ILD by alleviating vascular remodeling. This is attributed to the antiproliferative effect achieved via inhibition of TGFβ-related pathways on PASMCs. 

BLM-induced pulmonary vascular remodeling with thickening of media was attenuated in *Dpp4*KO mice in this study. Because pulmonary vascular remodeling is a hallmark of structural changes in PH, which comprise the abnormal proliferation of PASMCs and/or endothelial cells, pulmonary adventitial fibrosis, and inflammatory cell infiltration in the vascular walls, CD26/DPP4 could intervene somewhere in these mechanisms. This study focused on vascular SMC proliferation in PH-ILD models and clearly demonstrated in vitro that upregulated genes related to SMC proliferation and differentiation can be suppressed by *DPP4* knockdown. These findings suggest that media thickening in small pulmonary vessels was the main target in *Dpp4*KO mice. Treatment with sitagliptin, a DPP4 inhibitor, alleviates pulmonary artery remodeling in BLM-treated rats [28]. DPP4 inhibitors can decrease its enzymatic activity by binding to CD26/DPP4 on the surface of the lung constituent cells or the soluble form of DPP4 present in the circulating blood [29]. The downregulated CD26/DPP4 enzymatic activity in the targeting cells or in the circulating blood may contribute to the inhibition of pulmonary vascular remodeling in PH model animals. 

The degree of BLM-induced lung fibrosis in this study was similar between the two genotypes, whereas previous studies have shown that intratracheally administered BLM-induced lung fibrosis in mice was attenuated by *Dpp4* deficiency or treatment with the DPP4 inhibitor vildagliptin [22,30]. This discrepancy could be explained by the differences in BLM exposure methods: schedule, doses, and route of administration—especially intraperitoneal or intratracheal administration, which was reported to cause greater direct reactions in lungs [31]. The degree of involvement of *Dpp4*KO may be different between the development of PH and lung fibrosis induced by BLM administration. 

TGFβ production can be augmented in fibroblasts and macrophages by BLM challenge, which may play a central role in lung fibrosis and vascular remodeling [32]. To explore potential functional roles of CD26/DPP4 in PASMC in the BLM-induced PH model, *DPP4*-siRNA was used in cultured hPASMCs. The proliferation of hPASMCs was enhanced by TGFβ treatment, and this enhancement was suppressed by *DPP4*-siRNA (Figure 4d). Moreover, sitagliptin, a DPP4 inhibitor, inhibits the proliferation of hPASMCs induced by PDGF-BB [28]. PDGF-BB is a well-known potent mitogen implicated in proliferation and migration of PASMCs similar to TGFβ, playing a key role in the progression of PH. A reduction in DPP4 activity could suppress the proliferation of hPASMCs upon stimulation with cell growth factor. Moreover, the cytotoxicity assay revealed that LDH release from hPASMCs was not significantly changed by TGFβ treatment and was reduced by *DPP4*-siRNA treatment (Figure 4e). These results suggest that *DPP4*-siRNA can reduce cell damage in hPASMCs whereas the decrease in cell numbers, which was seen in proliferation assay, may also affect LDH concentration.

The molecular mechanisms, by which *DPP4*-siRNA suppresses SMC proliferation, were explored via transcriptome analysis of the cultured hPASMCs (Figure 5). The existence of CD26/DPP4 is essential for TGFβ receptor assembly [33]. This functional aid of CD26/DPP4 for the TGFβ receptors was observed in human microvascular endothelial cells and fibroblasts [34,35]. In this study, *TGFBR1* expression in hPASMCs was increased upon TGFβ stimulation, and this increase was suppressed by *DPP4*-siRNA treatment, indicating the existence of functional association of *DPP4* with *TGFBR1* in PASMCs. Moreover, transcriptome analysis showed that cell surface CD26/DPP4 could modulate intracellular TGF-β signaling via canonical and non-canonical pathways, leading to proliferation of PASMCs in PH pathobiology.

Enrichment analysis suggested that TGFβ signaling pathways including canonical and non-canonical pathways were downregulated by *DPP4*-siRNA treatment. The canonical pathway of TGFβ/Smad signaling could be related to the onset and development of PH [36], whereas Smad/Notch3 signaling activated by TGFβ stimulation promotes the proliferation of PASMCs via upregulation of SphK1/S1P [37]. Moreover, NOTCH3 overexpression in small pulmonary artery SMCs is a crucial signaling factor associated with the severity of PH in humans and rodents [38]. In the present study, *DPP4*-siRNA treatment downregulated TGFβ signaling possibly by interfering with canonical pathways, especially Notch3 pathways, and suppressed the proliferation of hPASMCs. On the contrary, TGFβ stimulation enhanced intracellular non-canonical pathways, which was suppressed by *DPP4*-siRNA treatment. TGFβ stimulation enhances non-canonical pathways including NFκB, RAF-MEK-ERK, p38 MAPK, JNK, and PI3K-Akt-mTOR [39]. Among these pathways, NFκB pathway is known to be associated with monocrotaline-induced PH by promoting vascular remodeling and increasing inflammation [40]. Furthermore, it has been suggested that the crosstalk between NFκB and Akt–mTOR signaling pathways may promote hypoxia-induced PH by increasing *DPP4* expression in PASMCs [27]. In this study, enrichment analysis on gene sets from hPASMCs demonstrated that the upregulation of PI3K-Akt and NFκB signaling by TGFβ stimulation was downregulated by *DPP4*-siRNA treatment.

This study has several limitations. First, we focused on the proliferation of PASMCs as a mechanism of vascular remodeling in this PH-ILD model based on the pathological findings of media thickening. However, various mechanisms, such as endothelial dysfunction, endothelial-to-mesenchymal transition, extracellular matrix production of fibroblasts, and release of inflammatory cytokines from macrophages, are related to vascular remodeling in patients with PH-ILD. Therefore, further evaluation is required to clarify the mechanisms underlying the in vivo roles of CD26/DPP4, including those in cell–cell interactions and cell transformation in endothelial cells, fibroblasts, and macrophages. Second, the molecular pathways related to hPASMC proliferation were explored using transcriptome analysis by RNA sequencing. However, verification of mRNA and protein expression should be performed using real-time quantitative PCR and/or Western blotting, and further independent confirmatory experiments to investigate molecular mechanisms are required. Third, it would be meaningful to explore if PASMC functions are different between patients with PH-ILD and healthy controls and are associated with CD26/DPP4 expression levels. Finally, it would be helpful to identify the substrates of CD26/DPP4 involved in pulmonary vascular remodeling and examine the effects of CD26/DPP4 activation on media component cells, including PASMCs or other cell types. Further studies are warranted to better understand the functional role of CD26/DPP4 in PH-ILD.

## 4. Materials and Methods

### 4.1. Animal Model of Pulmonary Hypertension with Interstitial Pneumonia

Five-to six-week-old male C57BL/6J mice (body weight: 18–20 g) were purchased from CLEA Japan, Inc. (Tokyo, Japan) and used as WT mice. *Dpp4*KO mice with a C57BL/6 background were provided by the Department of Therapy Development and Innovation for Immune Disorders and Cancers, Graduate School of Medicine, Juntendo University (Japan). All mice were housed in ventilated cages with microisolator lids and were kept at an ambient temperature of 22 °C and in a 12 h light-dark cycle. All experiments were conducted according to protocols approved by the Review Board for Animal Experiments of Chiba University (Japan). To establish the PH-ILD model, WT and *Dpp4*KO mice were intraperitoneally administered 0.035 mg/g of BLM (Nippon Kayaku Co., Ltd. Tokyo, Japan) or phosphate-buffered saline (PBS; 0.035 mg/g) twice weekly for 4 weeks as previously described [41]. In this experimental design, mice were assigned to one of the following four groups: WT/PBS, WT/BLM, *Dpp4*KO/PBS, and *Dpp4*KO/BLM. All mice were tested under anesthesia and euthanized on day 33. 

### 4.2. Hemodynamic Analysis

Pulmonary hemodynamics were assessed while the mice were under mild anesthesia induced using isoflurane (3% for induction, 1% for maintenance), and their body temperatures were maintained at 37 °C. Right heart catheterization was performed according to the manufacturer’s protocol [42], whilst the mice were maintained under spontaneous breathing. The mice were then placed in the supine position, and a small incision was made on the right side of the neck, where the right jugular vein was identified. A 1.4F microtip pressure catheter (SPR-671. Millar OEM Solutions. Houston, TX, USA) was advanced through the incision into the RV. Using a Power-Lab data acquisition system (AD Instruments. Dunedin, New Zealand), RVSP, CO, HR and Max dP/dt in RV were continuously monitored and recorded. After completion of the measurements, the mice were euthanized with 5% isoflurane and their hearts were removed. The RV free wall was carefully dissected from the left ventricle and septum (LV + S) and weighed to calculate RV/LV + S (Fulton index) as an indicator of RV hypertrophy.

### 4.3. Histological Analysis

After the mice were euthanized, the lungs were perfused via the right ventricle with PBS and fixed in 10% formalin after expansion of the lung tissues. The left lungs were cut sagitally into two sections, embedded in paraffin, sectioned (2 μm), and mounted on slides. The specimens were stained with Masson’s trichrome (MT). The severity of pulmonary fibrosis was semiquantitatively assessed according to the method proposed by Ashcroft using the mean of 10 fields (magnification, ×100) per mouse, as previously described [43]. The extent of muscularization in small pulmonary vessels (<100 μm diameter) was examined using Elastica van Gieson (EVG) and α-smooth muscle actin (α-SMA) staining. Vessels were identified as non-muscularized (no α-SMA staining), partially muscularized (α-SMA staining in the part of vessels), or fully muscularized (α-SMA staining in the whole circumference of the vessels), and then the distribution (%) of the three categories was calculated using a previously reported method [44]. The values obtained represent the mean for 30 vessels (magnification, ×100) per mouse. The RV free wall was also processed into 2 μm sections and stained with MT to evaluate fibrosis in the RV.

### 4.4. Cell Culture and Treatments of Small Interfering RNA and TGF-β1

hPASMCs were purchased from PromoCell (Heidelberg, Germany) and cultured in Smooth Muscle Cell Growth Medium 2 (PromoCell) supplemented with 10% fetal bovine serum. hPASMCs were incubated at 37 °C in a 5% CO_2_ incubator and used at passages 4–6 for all experiments. For small interfering RNA (siRNA) transfection, *DPP4* siRNA (Cat# 4392421, siRNA ID: s4255) and non-specific control siRNA (Cat# 4390843, Silencer™ Select Negative Control No. 1 siRNA) were purchased from Thermo Fisher Scientific (Waltham, MA, USA). Using the Lipofectamine™ RNAiMAX Transfection Reagent (Thermo Fisher Scientific), hPASMCs were transfected with siRNA for 48 h according to the manufacturer’s protocol. After the siRNA treatment, the cells were stimulated with recombinant human transforming growth factor-b1 (TGFβ) (PEPROTECH. Cranbury, NJ, USA) at a concentration of 10 ng/mL as previously reported [45] or with PBS at the same concentration for 24 h.

### 4.5. Proliferation and Cytotoxicity Assay

Cultured hPASMCs were treated with *DPP4* siRNA or control siRNA, detached using ACCUTASE (Thermo Fisher Scientific), and cultured in serum-free medium for 24 h and subsequently challenged with TGFβ or PBS. For the proliferation assay, the Cell Counting Kit-8 (WST-8. Dojindo Molecular Technologies. Kumamoto, Japan) was used according to the manufacturer’s protocol. In brief, the treated cells were added to a 96-well plate, 10 μL of WST-8 was added to each well, and the plate was incubated at 37 °C for 2 h. Cell viability was determined by measuring the absorbance at 450 nm using a microplate reader. For the cytotoxicity assay, a Cytotoxicity LDH Assay Kit-WST (Dojindo Molecular Technologies) was used to measure LDH expression levels according to the manufacturer’s protocol. The treated cells were added to a 6-well plate and 100 µL of LDH substrate solution was added to the wells, followed by incubation of the plates for 30 min at room temperature. The absorbance at 450 nm was measured using a microplate reader.

### 4.6. Real-Time Quantitative PCR Analysis

The total RNA was extracted from whole mouse lungs or cultured hPASMCs using the TRIzol reagent and the Direct-Zol RNA MiniPrep Plus Kit (ZYMO RESEARCH Corporation. Irvine, CA, USA). The extracted RNA was reverse transcribed via PCR using the SuperScript IV VILO Master Mix (Thermo Fisher Scientific) to synthesize single-stranded cDNA. cDNA samples both from mouse lungs and cultured hPASMCs were amplified using qPCR with the Fast SYBR Green PCR Master Mix (Thermo Fisher Scientific) and the GeneAmp PCR System (Thermo Fisher Scientific). Specific primers (the details of primer sequences are provided in Appendix A) were designed using an online software from the Universal Probe Library Assay Design Center (URL: https://lifescience.roche.com/en_us/brands/universal-probe-library.html#assay-design-center/) accessed on 13 June 2016 (Roche Applied Science. Upper Bavaria, Germany). The expression levels of target genes were normalized to hypoxanthine phosphoribosyl transferase 1 threshold cycle (CT) values and calculated using the 2^−∆∆Ct^ method (∆∆CT = [target gene CT of experimental group − reference gene CT of experimental group] − [target gene CT of control group − reference gene CT of control group]).

### 4.7. Flow Cytometry Analysis

In vivo, mouse lungs were perfused from the right ventricle until they were blood-free, using 20 mL of PBS containing 10 U/mL heparin (Mochida. Tokyo, Japan). The whole lungs were then minced and digested in an enzyme cocktail of Dulbecco’s modified Eagle’s medium (Sigma-Aldrich. Saint Louis, MO, USA) containing 1% bovine serum albumin (BSA) (Sigma), 2 mg/mL collagenase (Worthington. Lakewood, NJ, USA), 100 μg/mL DNase (Sigma), and 2.5 mg Dispase II (Sigma) at 37 °C for 60 min, followed by meshing through a 70 μm nylon cell strainer. The single cell suspensions were pretreated with an anti-CD16/32 antibody (BioLegend. San Diego, CA, USA) for 10 min to block Fc receptors, then incubated with specific antibodies in the dark at 4 °C for 15 min. The following antibodies were used for cell-surface staining: anti-CD26-PE, anti-CD31-PE/Cy7, and anti-CD45-Alexa Fluor 700 (BioLegend). After surface staining, the lung cells were fixed, permeabilized, and further incubated with anti-α-SMA (Thermo Fisher Scientific), followed by donkey anti-rabbit IgG-PE (Invitrogen. Boston, MA, USA) as the secondary antibody for 15 min in the dark at 4 °C. Cell fluorescence was measured with the BD FACS Canto™ II (BD Biosciences), and the data were analyzed using the FlowJo software ver. 10.8.1 (Becton, Dickinson and Company. Franklin Lakes, NJ, USA). To evaluate protein expression levels, the MFI of each sample was calculated (MFI = MFI of a sample stained with an antibody − MFI of an unstained sample [autofluorescence of the sample]). In vitro, cultured hPASMCs were pretreated with the anti CD16/32 antibody (BioLegend) for 10 min to block Fc receptors and then stained with anti-CD26-PE/Cy7 (BioLegend).

### 4.8. Transcriptome Analysis

Total RNA was isolated from the PASMCs and stored in Isogen (Nippon Gene. Tokyo, Japan). One milliliter of this solution was vigorously vortexed and then centrifuged after adding 200 µL of chloroform. The supernatants were removed, and 10 µg of glycogen (Roche. Basel, Switzerland) was added. RNA was precipitated by adding 500 µL of isopropyl alcohol. The solution was then vortexed vigorously and centrifuged. The RNA pellets were washed with 75% ethanol and then dissolved in 10 µL RNase-free water. The concentration and quality of RNA were verified using an Agilent 2100 Bioanalyzer (Agilent Technologies. Santa Clara, CA, USA). Purified total RNA (200 ng) with an RIN value > 9 was used for RNA library preparation according to the instructions of the QuantSeq 3′mRNA-Seq Library Prep Kit FWD for Illumina (Lexogen. Vienna, Austria). Libraries were amplified via 13 PCR cycles. RNA libraries were sequenced using an Illumina (San Diego, California, USA) NextSeq 500 system (75 cycles). The FASTQ files were prepared with reads using bcl2fastq ver2.20 (Illumina). The quality of FASTQ sequence data was assessed using FastQC v0.11.9 (Illumina). After removing adapter sequences from the raw reads, the trimmed reads were aligned using STAR v2.7.6a to the GRCh38 human reference genome. Reads per million values were calculated using Samtools v1.15, and htseq count v1.99.2. The expression levels of the genes identified in the transcriptome were normalized and compared. Principal component analysis and heat maps with hierarchical clustering were created using the Qlucore Omics Exploration software ver. 3.9.9 (Qlucore AB. Lund, Sweden). The fold change between each group was >4.0 (*p* < 0.001). Significantly over-represented functional categories were identified using Enrichr online tool (http://amp.pharm.mssm.edu/Enrichr/) accessed on 1 October 2023. Genes with significantly upregulated expression between Control/PBS and Control/TGFβ or downregulated expression between Control/TGFβ and *DPP4*KD/TGFβb were annotated. GO terms and KEGG pathways were also identified and considered significant at *p* < 0.05.

### 4.9. Statistical Analysis

Data are presented as the mean ± standard deviation. Unpaired two-tailed *t*-tests were used for comparisons of two groups. One-way ANOVA was used for multiple group comparisons, followed by Bonferroni’s post hoc test. Statistical significance was set at *p* < 0.05. Statistical analyses were performed using GraphPad Prism version 9.3 (GraphPad Software, Inc. San Diego, CA, USA).

## 5. Conclusions

This study demonstrated that genetic deficiency of *Dpp4* has protective effects on BLM-induced PH in mice by alleviating vascular remodeling, potentially by exerting an antiproliferative effect on PASMCs via the Notch, PI3K-Akt, and NFκB signaling pathways. Therefore, CD26/DPP4 may be a potential therapeutic target in patients with PH associated with ILDs.

## Figures and Tables

**Figure 1 ijms-25-00748-f001:**
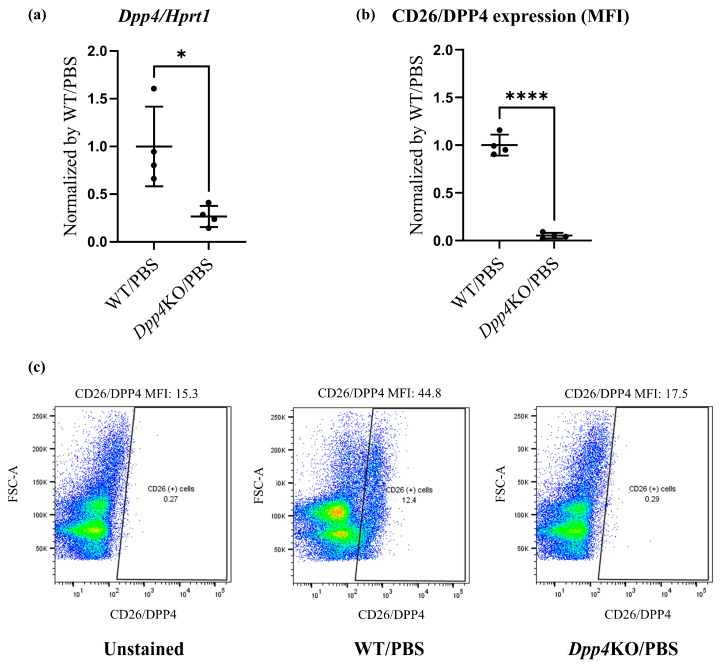
Pulmonary hemodynamic evaluation of bleomycin (BLM)-induced pulmonary hypertension (PH) in wild type (WT) and *Dpp4*KO mice. CD26/DPP4 expression in whole lung cells of WT and *Dpp4*KO mice was measured using (**a**) real-time quantitative PCR and (**b**) flow cytometry. Panel (**c**) shows representative images of dotted plots. The pulmonary hemodynamic parameters evaluated by right heart catheterization were as follows: (**d**) right ventricular (RV) systolic pressure (RVSP), (**e**) heart rate (HR), (**f**) cardiac output (CO), and (**g**) maximal rate of pressure rise (max dP/dt) in RV. RV hypertrophy was evaluated by calculating (**h**) the Fulton index (weight ratio of the right ventricle to the left ventricle plus the ventricular septum) or (**i**) RV/body. ns, not significant; * *p* < 0.05, ** *p* < 0.01, *** *p* < 0.001, **** *p* < 0.0001.

**Figure 2 ijms-25-00748-f002:**
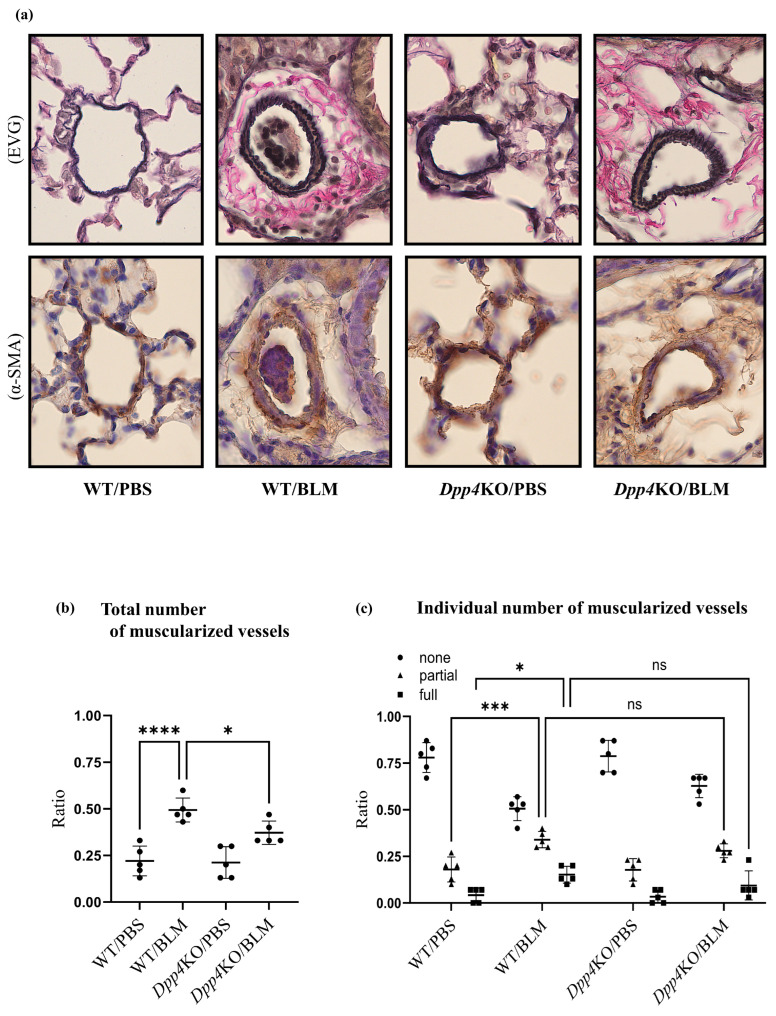
Evaluation of pulmonary small vessel remodeling of BLM-induced pulmonary hypertension in WT and *Dpp4*KO mice. The representative images of small pulmonary vessels of four groups are shown in (**a**): staining with Elastica van Gieson (EVG) and α-SMA; observed under ×400 magnification. The extent of vascular muscularization (*n* = 5 in each group) is summarized in (**b**): the total number of partially and fully muscularized vessels, and in (**c**): individual number of muscularized vessels. The mean fluorescence intensity of α-SMA in CD31^+^CD45^−^ pulmonary vascular endothelial cells and whole lung cells is shown in (**d**,**e**). Representative dot plot images of CD31^+^CD45^−^ pulmonary vascular endothelial cells are shown in panel (**f**). The representative images of four groups are shown in (**g**): lung tissues (Masson’s trichrome (MT); ×40 magnification), and in (**h**): right ventricle free wall (Masson’s trichrome; ×40 magnification). To evaluate lung fibrosis, the Ashcroft score (*n* = 5 for each group) was calculated (**i**). ns, not significant; * *p* < 0.05, *** *p* < 0.001, **** *p* < 0.0001.

**Figure 3 ijms-25-00748-f003:**
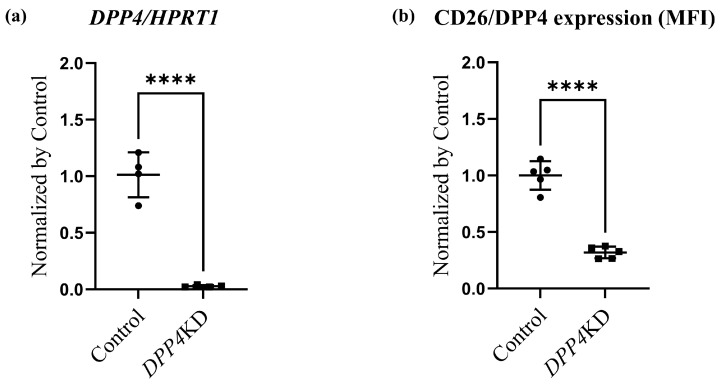
Cell viability and damage in cultured hPASMCs were reduced by *DPP4* knockdown. Expression levels of CD26/DPP4 in cultured hPASMCs were evaluated using real-time quantitative PCR (**a**) and flow cytometry (**b**). Representative dot plots of flow cytometry are shown in panel (**c**). Cell viability and cytotoxicity were evaluated using (**d**): Cell Counting Kit-8 assay and (**e**): LDH assay, respectively. Control: hPASMCs treated with nonspecific control siRNA; *DPP4* knockdown (*DPP4*KD): hPASMCs treated with *DPP4*-siRNA; ns: not significant; ** *p* < 0.01, *** *p* < 0.001, **** *p* < 0.0001.

**Figure 4 ijms-25-00748-f004:**
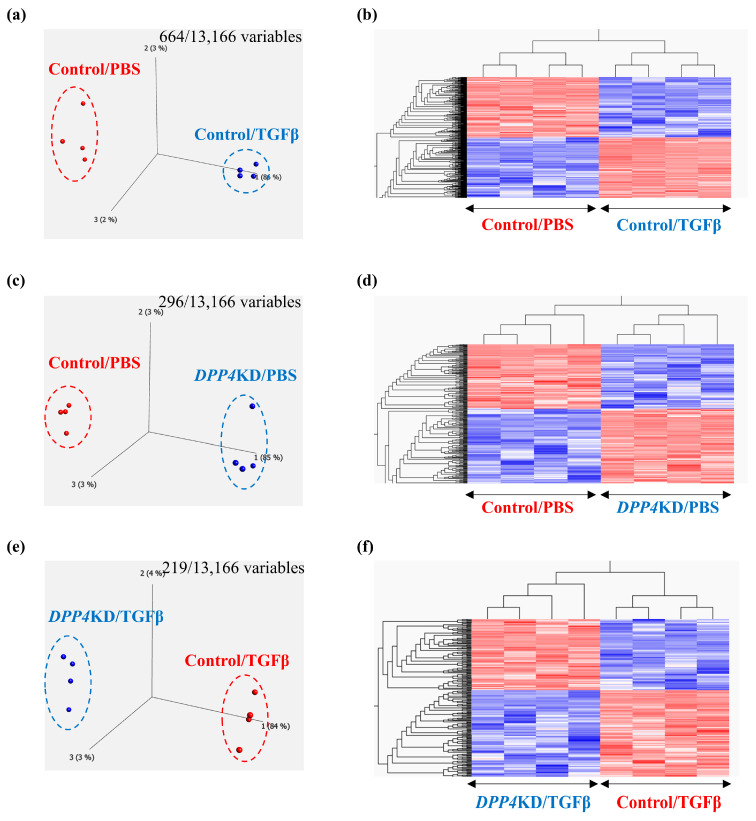
Transcriptome analysis of cultured hPASMCs treated with TGFβ and *DPP4*-siRNA. Cultured hPASMCs were treated as follows (each group: *n* = 4): Control/PBS (treated with non-specific control siRNA followed by PBS), Control/TGFβ (treated with non-specific control siRNA followed by TGFβ treatment), *DPP4*KD/PBS (hPASMCs treated with *DPP4*-siRNA followed by PBS), *DPP4*KD/TGFβ (hPASMCs treated with *DPP4*-siRNA followed by TGFβ treatment). Principal component analysis and heat map with hierarchical clustering of differentially expressed genes between the Control/PBS and Control/TGFβ (**a**,**b**), Control/PBS and *DPP4*KD/PBS (**c**,**d**), and Control/TGFβ and *DPP4*KD/TGFβ groups (**e**,**f**) are shown. mRNA expression levels of TGFβ pathway-related genes in hPASMCs are shown (**g**–**r**). ns; not significant, * *p* < 0.05, ** *p* < 0.01, *** *p* < 0.001, **** *p* < 0.0001.

**Figure 5 ijms-25-00748-f005:**
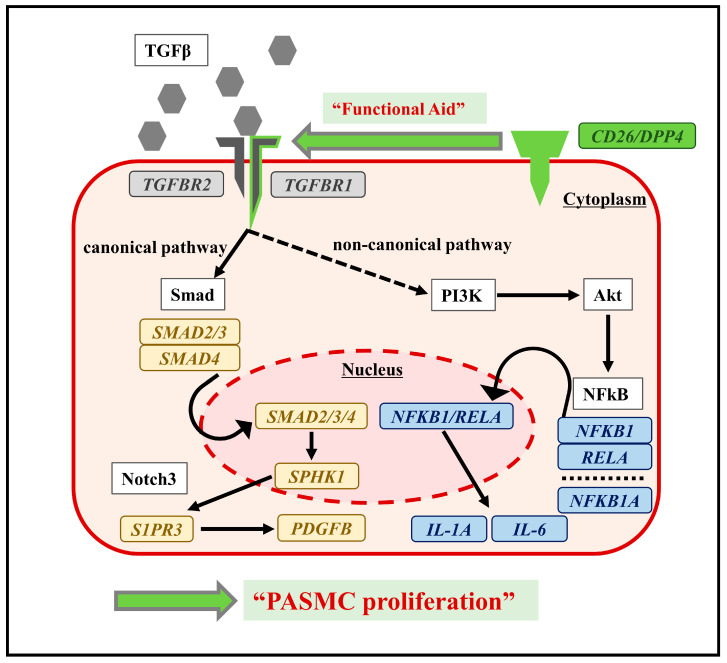
Conceptual diagram of functional association of CD26/DPP4 with TGFβ via signaling pathways in hPASMCs. Potential mechanisms of PASMC proliferation are also shown. Transcriptome analysis of hPASMCs suggested that the functional aid of CD26/DPP4 for TGFβ receptors might activate canonical and non-canonical pathways, causing PASMC proliferation.

**Table 1 ijms-25-00748-t001:** Enrichment analysis of transcriptomic data (Control/PBS vs. Control/TGFβ).

(A): GO: relevant terms were excerpted	
Terms with upregulated genes following TGFβ treatment	*p*-value
Regulation of SMC differentiation (GO: 0051150)	0.0041
Cellular response to growth factor stimulus (GO: 0071363)	0.0051
Regulation of vascular associated SMC migration (GO: 1904754)	0.0057
Regulation of SMC proliferation (GO: 0048660)	0.013
Pathway-restricted SMAD protein phosphorylation (GO: 0060393)	0.020
Response to TGF-beta (GO: 0071559)	0.026
Regulation of TGF-beta production (GO: 0071634)	0.030
Notch signaling pathway (GO: 0007219)	0.030
Regulation of TGF-beta receptor signaling pathway (GO: 0017015)	0.041
Regulation of vascular associated SMC proliferation (GO: 1904707)	0.045
(B): KEGG: relevant pathways were excerpted	
Pathways with upregulated genes following TGFβ treatment	*p*-value
PI3K-Akt signaling pathway	<0.0001
Cytokine–cytokine receptor interaction	0.013
Notch signaling pathway	0.018
TGF-beta signaling pathway	0.076
NF-kappa B signaling pathway	0.10

**Table 2 ijms-25-00748-t002:** Enrichment analysis of transcriptomic data (Control/TGFβ vs. *DPP4*KD/TGFβ).

(A): GO: relevant terms were excerpted	
Terms with downregulated genes following *DPP4*-siRNA treatment	*p*-value
Cellular response to growth factor stimulus (GO: 0071363)	<0.0001
Cellular response to cytokine stimulus (GO: 0071345)	<0.0001
Notch signaling pathway (GO: 0007219)	0.0018
Pathway-restricted SMAD protein phosphorylation (GO: 0060389)	0.0031
SMAD protein signal transduction (GO: 0060395)	0.0036
Regulation of TGF-beta receptor signaling pathway (GO: 0017015)	0.029
Regulation of vascular associated SMC differentiation (GO: 1905063)	0.035
Positive regulation of NIK/NF-kappa B signaling (GO: 1901224)	0.040
Regulation of vascular associated SMC migration (GO: 1904754)	0.041
Regulation of SMC proliferation (GO: 0048660)	0.042
(B): KEGG: relevant pathways were excerpted	
Pathways with downregulated genes following *DPP4*-siRNA treatment	*p*-value
Cytokine–cytokine receptor interaction	0.0020
PI3K-Akt signaling pathway	0.020
NF-kappa B signaling pathway	0.0036
TGF-beta signaling pathway	0.019
Notch signaling pathway	0.049

**Table 3 ijms-25-00748-t003:** Differences in gene expression levels in hPASMCs.

(A): Genes related to *DPP4* and the *TGF*β family					
Gene ID	Control /PBS	Control /TGFβ	*DPP4*KD /PBS	*DPP4*KD /TGFβ	*p*-value
*DPP4*	43.50	21.74	0.37	0.81	<0.0001
*TGFB1*	55.25	80.83	42.85	82.31	<0.0001
*TGFBR1*	25.78	77.10	28.13	54.63	<0.0001
*TGFBR2*	116.50	55.40	98.17	52.79	<0.0001
(B): Genes related to the canonical pathway (Notch pathway)					
Gene ID	Control /PBS	Control /TGFβ	*DPP4*KD /PBS	*DPP4*KD /TGFβ	*p*-value
*SMAD2*	52.51	39.64	46.69	44.30	0.064
*SMAD4*	16.08	20.70	11.76	11.93	0.0003
*SPHK1*	49.69	98.18	44.09	73.12	<0.0001
*S1PR3*	38.57	113.9	33.89	74.95	<0.0001
*NOTCH3*	21.81	59.39	24.31	42.56	<0.0001
*PDGFB*	6.55	21.10	2.59	8.01	<0.0001
(C): Genes related to the non-canonical pathway (NFκB pathway)					
Gene ID	Control /PBS	Control /TGFβ	*DPP4*KD /PBS	*DPP4*KD /TGFβ	*p*-value
*NFKB1*	22.19	38.16	19.97	27.27	0.0077
*RELA*	44.16	49.64	46.41	50.99	0.56
*NFKB1A*	170.20	170.00	126.10	109.50	<0.0001
*IL-1A*	38.41	46.95	20.57	18.31	0.0003
*IL-6*	44.71	134.60	44.34	94.56	<0.0001
*CXCL8*	939.60	821.60	518.10	360.10	<0.0001

## Data Availability

The datasets presented in this study can be found online in the NCBI database (accession number: GSE248794).

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
