# Peer review of "Functional Roles of CD26/DPP4 in Bleomycin-Induced Pulmonary Hypertension Associated with Interstitial Lung Disease"

_ijms, 2024, doi:10.3390/ijms25020748_

Round 1

Reviewer 1 Report

Comments and Suggestions for Authors

First of all, the Reviewer would like to congratulate the authors for the current interesting study, which could give many researchers and clinicians useful information for the issue. The reviewer would like to raise some comments. However, the reviewer sincerely hope that those would be helpful for the authors.

1. Was there difference in the mortality between the WT/BLM mice and the DPP4ko/BLM nice?

2. The authors described in the discussin section that "TGFβ production can be augmented in fibroblasts and macrophages by BLM chalenge, which may play a central role in lung fibrosis and vascular remodeling". Do the authors think about the TGF-β-mediated roles of fibroblast and macrophage in actual tissues in the WT/BLM and the DPP4ko/BLM mice in the vitro study? Have you evaluated fiboblast and macrophage in WT/BML mice and DPP4ko/BML mice in actual tissues? 

3. What do the authors think about results from the in vitro study if using hPASMCs cultured from the lung of FLD-PH patients? Is there a difference?

Reviewer 2 Report

Comments and Suggestions for Authors

The authors demonstrated that genetic deficiency of Dpp4 protects against BLM-induced PH. The deficiency of Dpp4 reduces vascular remodeling, potentially through the exertion of an antiproliferative effect via inhibition of the TGFβ-related pathways in PASMCs.

The manuscript is well-written and easy to read. 

However, there are some method and results that are not entirely clear.

Methods: 

Do the authors use a live/dead probe after selecting the population in FC analyses?

Which section of the lung did the authors use for the RT-qPCR and IHC analyses? Add to methods. 

Please add to the methods a table with primer sequence and TM for each primer used.

Figure 1A: The authors showed images of IHQ indicating that the vascular remodelling was decreased. Please measure the medial layer and determined this hypothesis with data. There are many methods to determined the remodelling in histological images. 

Figure 2h: The authors showed only a representative dot plot indicating the MFI. Please graph and statistically compare the SMA MFIs.

Figure 3C: In FC dot plots, please graph the percentage and the MFI. 

Comments on the Quality of English Language

Fine

Round 2

Reviewer 1 Report

Comments and Suggestions for Authors

The revised manuscript has been sufficiently improved. The reviewer has no more comments for the revised manuscript.

Reviewer 2 Report

Comments and Suggestions for Authors

Accepted in present form.